# Simple and Robust Deep Learning Approach for Fast Fluorescence Lifetime Imaging

**DOI:** 10.3390/s22197293

**Published:** 2022-09-26

**Authors:** Quan Wang, Yahui Li, Dong Xiao, Zhenya Zang, Zi’ao Jiao, Yu Chen, David Day Uei Li

**Affiliations:** 1Department of Biomedical Engineering, University of Strathclyde, Glasgow G4 0RU, UK; 2Key Laboratory of Ultra-Fast Photoelectric Diagnostics Technology, Xi’an Institute of Optics and Precision Mechanics, Xi’an 710049, China; 3Department of Physics, University of Strathclyde, Glasgow G4 0NG, UK

**Keywords:** fluorescence lifetime imaging (FLIM), deep learning, imaging analysis

## Abstract

Fluorescence lifetime imaging (FLIM) is a powerful tool that provides unique quantitative information for biomedical research. In this study, we propose a multi-layer-perceptron-based mixer (MLP-Mixer) deep learning (DL) algorithm named FLIM-MLP-Mixer for fast and robust FLIM analysis. The FLIM-MLP-Mixer has a simple network architecture yet a powerful learning ability from data. Compared with the traditional fitting and previously reported DL methods, the FLIM-MLP-Mixer shows superior performance in terms of accuracy and calculation speed, which has been validated using both synthetic and experimental data. All results indicate that our proposed method is well suited for accurately estimating lifetime parameters from measured fluorescence histograms, and it has great potential in various real-time FLIM applications.

## 1. Introduction

Fluorescence lifetime imaging (FLIM) is a powerful technique in biomedical research [1,2], such as probing cellular microenvironments and physiological parameters, including pH, viscosity, temperature, and ion concentrations (e.g., Ca^2+^, O_2_) [3,4,5,6,7]. In combination with Förster resonance energy transfer (FRET), FLIM–FRET has been widely used as a “quantum ruler” to quantify protein–protein conformations and interactions [8,9]. Fluorescence lifetimes are local properties of fluorophores depending only on the physicochemical state of the local microenvironment (e.g., pH, ionic strength). They are free of artifacts due to fluctuations in laser power, optical path, and fluorophore concentrations. There are time and frequency domain approaches for measuring fluorescence lifetimes [10,11]. Among them, time-correlated single-photon counting (TCSPC) [12] has been widely used due to its superior photon efficiency, high signal-to-noise ratio, and superior temporal resolution [13]. TCSPC records photon arrival times after shining laser pulses and yields decay histograms, from which fluorescence lifetimes can be extracted. Theoretically, a fluorescence decay histogram can be expressed as [14]:(1)$$y\left(t\right)=\sum_{i=1}^{n}\alpha_{i}\exp\left(-\frac{t}{\tau_{i}}\right),\;\;\;\sum_{i=1}^{n}\alpha_{i}=1,$$
where *n* is the number of components, and $\alpha_{i}$ and $\tau_{i}$ are the fraction and the lifetime of the *i*-th component. Curve-fitting methods have been commonly used for FLIM analysis, including the nonlinear least-square method (NLSM) [15], maximum likelihood estimation (MLE) [16], Bayesian analysis [17], and Laguerre expansion methods [18]. However, they require many photons to deliver accurate results. They are time-consuming and limited to offline analysis. Moreover, they require prior knowledge to set proper initial parameters. Fitting-free methods have also been developed, such as the phasor approach [19] and the center-of-mass method (CMM) [20]. The phasor approach offers a fast graphic interpretation [21,22]. CMM [20,23] offers fast analysis, but it can only provide intensity-weighted average lifetimes [14].

Deep learning (DL) techniques have been proven promising for FLIM analysis [24,25]. It is to extract high dimensional features of input decay histograms and then map the input to the lifetime parameters. In contrast, the Bayesian estimation is to estimate the posterior distribution from the chosen prior distribution and likelihood function. Usually, the posterior distribution can be obtained analytically or approximately using Markov chain Monte Carlo (MCMC) methods. A general structure of a deep-learning-based predictive model is shown in Figure 1a, including the model training from training data in phase 1 and model testing from new data in phase 2. Various neural networks were proposed for multiexponential analysis, including multilayer perceptions (MLPs) [26], one- and high-dimensional convolutional neural networks (CNNs) [27], online training extreme learning machine [28], and generative adversarial networks (GANs) [29]. MLP is simple and can be easily implemented on different platforms, such as CPU, GPU, or other embedded systems. According to the general approximation theorem, MLP can approximate any continuous function. However, it shows poor feature extraction capacity. Therefore, MLP-based algorithms cannot resolve multiexponential decay models [26]. A feasible way to address this problem is adding feature engineering as network input. For example, phasor coordinates can be considered to improve the resolvability of multiexponential analysis. CNN is powerful for feature extraction, such as local parameter sharing, sparse interactions, and equivariant representations. High-dimensional CNNs have been successfully applied for multiexponential FLIM analysis with high precision [27]. Meanwhile, the light-weighted one-dimensional (1D) CNN has been proposed for FLIM analysis [30]. The 1D-CNN features high efficiency, fast training, and high inference speed. It is also hardware friendly to be applied in embedded systems. However, CNN heavily relies on convolution operators and requires hyperparameter optimization, and its performance is significantly affected by the size of convolutional kernels, because CNNs only have a local field-of-view with a convolutional kernel, which cannot simultaneously learn the complete decay information.

We proposed a new MLP-based DL algorithm to address the above challenges. Our algorithm adopts the MLP-Mixer [31] to provide fast and accurate analysis even under a low signal-to-noise ratio (SNR). Inspired by CNNs and transformers, the MLP-Mixer has an isotropic design with only MLP layers, which is easy to implement. Thus, our algorithm is particularly suitable for embedded systems. Compared with CNNs, the MLP-Mixer has three distinct advantages. First, it has higher computational efficiency because only matrix multiplications are involved. Second, the MLP-Mixer is more suitable for analyzing sequence signals. It has a global field of view, processing the whole decay sequence at the same time. Third, the implementation and optimization of the MLP-Mixer are simple and suitable for broad applications. We demonstrate that the MLP-Mixer is efficient, robustly solving simulated and experimental FLIM data. We compared the proposed method with 1D-CNN, NLSM, and VPM in terms of accuracy and inference time for one FLIM image; see Figure 1b for an intuitive summary. The results indicate that the MLP-Mixer outperforms traditional methods with less bias, especially when the fluorescent samples have small lifetime components. The inference time for one FLIM image is around 163-fold and 900-fold faster than NLSM and the variable projection method (VPM) [32], respectively.

## 2. Methods and Theory

### 2.1. Architecture Design

Figure 2a shows the topological structure of the proposed the FLIM-MLP-Mixer. The input histogram is firs divided into nonoverlapping patches. The patch size is set to 10 for capturing more general features of decay curves while maintaining a moderate temporal resolution. All patches are fed into a mixer block simultaneously. The core module of the network is the mixer, which consists of layers that mix features (i) at a given spatial location and (ii) between different spatial locations. For the mixer design, there are two MLP blocks. One is the token-mixing MLP, in which an input (batches × patches × channels) is accepted by a norm layer and then transposed to a matrix (batches × channels × patches). The data are then passed through the MLP1 block, which consists of two fully connected layers with a nonlinear activate function (GELU) [33]. The output of the MLP1 block is transposed to a matrix (batches × patches × channels) for the next MLP block, which is the channel-mixing MLP block. This MLP block (MLP2) is responsible for extracting patch features. In either token-mixing or channel-mixing MLP block, skip connections [34] are employed. After the mixer, there are three dense layers for reconstructing average lifetimes. The exhaustive searching method is used to optimize the hyperparameters of the neural networks.

The idea behind the mixer is to separate the per-location (channel-mixing) operations and cross-location (token-mixing) operations. The skip connections can avoid gradient vanishing problems, leading to faster convergence and better performance. Our network has advantages with a light-weighted structure. The mixer relies on basic matrix multiplication routines, which require fewer hardware resources, including smaller memory size and less power consumption. It is much faster without sophisticated matrix operations. We can quickly implement hardware-friendly MLP neural networks in field-programmable gate array (FPGA) devices and integrate them into a wide-field SPAD FLIM system. In the training process, we use one block for the MLP-Mixer, in which the dimension of the token and channels are 16, and the patch size is 10.

### 2.2. Training Dataset Preparation

Training data are critical for neural networks. FLIM training data can be easily obtained using synthetic data since mathematical FLIM decay and noise models are well developed. A theoretical fluorescence signal measured by a FLIM system can be described as [30]:(2)yt=IRFt∗A∑i=1nαie−tτi+εt,  ∑i=1nαi=1,
where *A* is the amplitude of the underlying fluorescence decay, $\alpha_{i}$ is the *i*-th fraction ratio, $\tau_{i}$ is the *i*-th lifetime component, and $\varepsilon \left(t\right)$ is the Poisson noise, which is dominant in a TCSPC system. *IRF*(**·**) is the FLIM instrument response function, approximated by a Gaussian function with FWHM = 167 ps. The asterisk (∗) refers to a convolution operator. The total photon counts $Y=\int y\left(t\right)dt$ of synthetic signals range from 100 to 1 × 10^4^. The SNR of a signal is defined as:(3)$$\mathrm{SNR}\;=20*\log\sqrt{Y},$$

In most FLIM applications, signals’ underlying decay model is unknown; it is often valuable for determining average lifetimes rather than lifetime components, which are suspicious for biological interpretation. Amplitude-weighted average lifetimes can estimate FRET efficiency and access dynamic quenching behaviors, which are of great interest [23]. Therefore, the FLIM-MLP-Mixer is designed for evaluating amplitude-weighted average lifetimes, defined as [14]:(4)$$\tau_{A}=\sum_{i=1}^{n}\alpha_{i}\tau_{i},$$

### 2.3. Training and Evaluation

The proposed neural network was deployed in PyTorch 1.0 [35]. The loss function is defined as:(5)$$L\left(\Theta \right)=\frac{1}{N}\sum_{i=1}^{N}{\left\|F\left(X^{i},\Theta \right)-Y^{i}\right\|}^{2},$$
where *X* is the input and *Y* is the corresponding label of the signal. *F* is the mapping function with the network parameters *Θ*, and *N* is the batch size of the training dataset. The optimizer uses the Adam algorithm with a learning rate of 1 × 10^−4^ in the standard back-propagation [36]. The training dataset contains 40,000 different signals and their corresponding lifetimes. Both mono- and biexponential decays were included in training datasets. For monoexponential signals, the lifetime range is [0.1, 5] ns, and for biexponential signals, $\tau_{1}\;and\;\tau_{2}$ are set in [0.1, 1], [1, 5] ns. The trained lifetimes cover a wide range of lifetimes of commonly used fluorophores for biomedical applications. Figure 2b shows the signals under different SNRs, the green curve stands for decay, and the red curve means IRF.

The FWHM of IRFs has a Gaussian distribution with a mean value of 167 ps and a standard deviation of 60 ps. The training batch size is 200, and 20% of the total datasets are used for validation. An early stop callback with 20 patients is included to prevent overfitting. All the datasets are normalized to 1 before being fed into the neural network. Figure 2c shows the training and validation errors of the MLP-Mixer. The mean square errors (MSEs) of both training and validation losses decrease rapidly and reach the plateau after 120 epochs. They converge at a small value of 1.8 × 10^−4^, indicating that the network is well trained as the estimated lifetimes are close to their ground truths. Because of the simple network structure, the number of hyperparameters is only 26,081. The training time is half an hour on a typical desktop using a central processing unit (CPU).

## 3. Simulation Results

The MLP-Mixer was first evaluated on synthetic monoexponential signals with lifetimes of 1 and 4 ns. The photon counts range from 100 to 10,000, with 20 to 40 dB SNRs. We define the relative bias as Bias = |$\tau_{pre}-\tau_{truth}$|/$\tau_{truth}$, where $\tau_{pre}$ and $\tau_{truth}$ are the estimated $\tau_{A}$ by the methods and label of $\tau_{A}$ from Equation (4), respectively. We define the standard deviation, Std=∑i=1nτAi−τtruth2n−1, where $\tau_{A}^{i}$ is the predicted parameter, and *n* is the number of simulated decay curves. Figure 3a,b shows the violin plots of the predicted results of the simulated data. The red dotted line stands for the ground truth of lifetimes. The result of the MLP-Mixer is closer to the ground truth, reaching 0.95, 0.96, and 0.98 ns at SNRs 20~32 dB, 32~38 dB, and 38~40 dB, respectively, for signals with a lifetime of 1 ns. Figure 3c,d is the corresponding bias and standard deviation plots. Both figures show that the MLP-Mixer and 1D-CNN outperform the traditional methods NLSM and VPM in terms of accuracy under different SNRs for both short and long lifetimes. The biases of the estimation with the MLP-Mixer are lower than 2% and 5% for short and long lifetimes, meaning that the MLP-Mixer is a more accurate estimator. In addition, as shown in Figure 3c, the standard deviations of all methods for short lifetimes are less than 0.06, which is much smaller than those shown in Figure 3d, which are above 0.1, indicating that methods for short lifetimes are more precise than long lifetimes under the same photon counts and SNRs. This is because, in TCSPC, the noise is governed by Poisson distribution, so the SNR is proportional to the square root of total photon counts. For a given photon count, for example, 2000 p.c., a larger lifetime means that photons are distributed in more time bins. Therefore, each time bin has fewer photons, and the SNR is smaller, so larger lifetimes yield a more significant deviation. In conclusion, the MLP-Mixer is more accurate than 1D-CNN, NLSM, and VPM with acceptable precision under different SNRs.

We also evaluate our method on histograms following a biexponential decay model. The short and long lifetime components, $\tau_{1}$ and $\tau_{2}$, range from 0.5 to 1.5 ns and from 2.5 to 3.5 ns, respectively. α equals 0.2, 0.5, and 0.8. We compared the inference performances of the MLP-Mixer, 1D-CNN, NLSM, and VPM for amplitude-weighted lifetimes. In this simulation, 2000 simulated testing datasets were generated using Equation (4). Figure 4 shows the results for SNR 26 dB, 34 dB, and 38 dB and α ranging from 0.2 to 0.8. From Figure 4a,d,g, we can see that the MLP-Mixer has a low bias of 0.037, 0.057, and 0.006 at α = 0.2, 0.5, and 0.8, respectively, and NLSM and VPM have bias over 0.1 under the same conditions, indicating that the MLP-Mixer outperforms NLSM and VPM. It is obvious that the bias of the MLP-Mixer at α = 0.8 is much smaller than that of 1D-CNN, NLSM, and VPM at α = 0.2, 0.5, because larger α means the short lifetime components in the decay are dominant, which have a good agreement with the discussion in Figure 3. Moreover, NLSM and VPM are more sensitive to SNRs than the MLP-Mixer and 1D-CNN. However, when α = 0.5, the MLP-Mixer and 1D-CNN have a very similar result.

Table 1 summarizes the main characteristic parameters of the MLP-Mixer and 1D-CNN. The number of hyperparameters of the MLP-Mixer is threefold more significant than that of 1D-CNN, whereas the training time of the MLP-Mixer (1871.0 s) is slightly less than that of 1D-CNN (2270.62 s). The total numbers of floating-point operations (FLOPs) for a single sample are very similar, 5.9 billion and 5.4 billion for the MLP-Mixer and 1D-CNN, respectively. The inference time of 1D-CNN is 34.06 s, whereas the MLP-Mixer only consumed 12.54 s, showing that the efficiency of the MLP-Mixer is better than 1D-CNN.

## 4. Experimental Results and Discussion

Dye solutions and plant cell samples were used for experimental validation. FLIM datasets were obtained with a commercial two-photon FLIM system.

### 4.1. Rhodamine 6G and Rhodamine B Solutions

Rhodamine B and 6G saturated aqueous solutions were placed on a glass slide with coverslips. The FLIM system consists of a confocal microscope (LSM 510, Carl Zeiss, Oberkochen, Germany) and a TCSPC card (SPC-830, Becker & Hickl GmbH, Berlin, Germany). The femtosecond Ti: sapphire laser (Chameleon, Coherent, Santa Clara, CA, USA) has an excitation wavelength of 800 nm and a pulse width of less than 200 fs with an 80 MHz repetition rate. The bin width of TCSPC is 0.039 ns, and each measured histogram contains 256-time bins. The emission light passes through a 500–550 nm bandpass filter and then is collected by a 60× water-immersion objective lens (N.A = 1.0). We experimented on rhodamine 6G and rhodamine B at different SNRs to mimic low-light conditions. The results are shown in Table 2.

Table 2 summarizes the results calculated by the MLP-Mixer, NLSM, VPM, and 1D-CNN. τ_REF is the reference lifetime of the tested samples (rhodamine 6G and rhodamine B) [37,38]. In our study, a threshold (10% of total counts) was applied to mask pixels with low photon counts. The MLP-Mixer and 1D-CNN outperform NLSM and VPM in terms of accuracy. Results show that the MLP-Mixer is better than others, even under a low SNR. That is because a lower SNR means a lower photon count, which is not enough for fitting methods.

### 4.2. Convallaria majalis Cells

We further evaluated the MLP-Mixer on *Convallaria majalis* cell samples. The sample was measured with ‘high photon counts’ (HPC) at a 30 s acquisition time, ‘middle photon counts’ (MPC) at a 15 s acquisition time, and ‘low photon counts’ (LPC) at a 3 s acquisition time. Figure 5 shows the results for three cases. LPC data analysis is challenging for NLSM due to the low SNR. The performances of these results were evaluated via the structural similarity index (SSIM) [39], summarized in Table 3. The MLP-Mixer significantly outperforms NLSM, especially in low SNR situations, where the SSIM values are 0.95 and 0.81 for the MLP-Mixer and NLSM, respectively. From the lifetime images, we can see that Figure 5e,i is very similar to Figure 5h,l, indicating that the MLP-Mixer and 1D-CNN performed well even at low counts. That is because both DL algorithms extract the features of decay histograms in high dimensional space, making the algorithms insensitive to different SNR levels. In Figure 5x, the lifetime distributions show that all methods obtain similar results at a high SNR.

## 5. Conclusions

In summary, we present the design and training of the MLP-Mixer for FLIM analysis. The proposed algorithm has a simple architecture, which is easy to implement and hardware friendly. The simulated FLIM data analysis shows that the MLP-Mixer performs better, especially for small lifetime components. The trained model was then employed to analyze two-photon FLIM images of rhodamine 6G, rhodamine B, and *Convallaria majalis* cells. The results also suggest the MLP-Mixer’s excellent performance and robustness. Due to its simple architecture, it has the potential to be implemented in FPGA devices to accelerate the analysis for real-time FLIM applications.

## Figures and Tables

**Figure 1 sensors-22-07293-f001:**
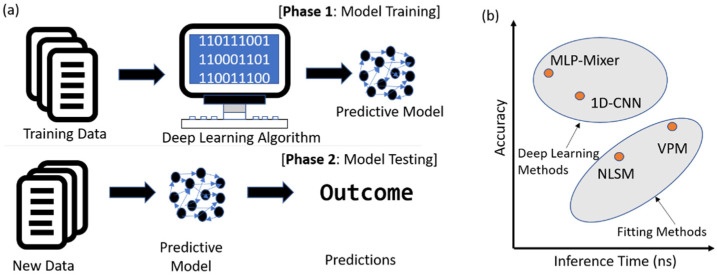
(**a**) Logic of the deep learning algorithm based on a predictive model considering training and testing phases. (**b**) Comparison of fluorescence lifetime prediction methods related to inference time and accuracy for one FLIM image.

**Figure 2 sensors-22-07293-f002:**
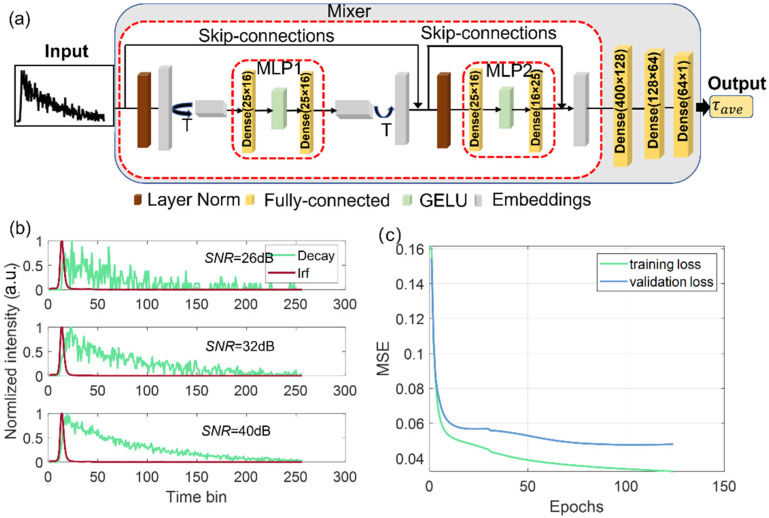
(**a**) MLP-Mixer architecture. The MLP-Mixer consists of per-patch linear embeddings, mixer layers, and a regression layer. The mixer block contains one token-mixing MLP (MLP1) and one channel-mixing MLP (MLP2), each consisting of two fully connected layers and a GELU nonlinearity. Other components include skip connections, dropout, and layer norm on channels. (**b**) Decays under different SNRs, the green curve stands for decay, and the red curve means IRF. (**c**) Mean square error (MSE) plots of the training and validation loss for τ_ave.

**Figure 3 sensors-22-07293-f003:**
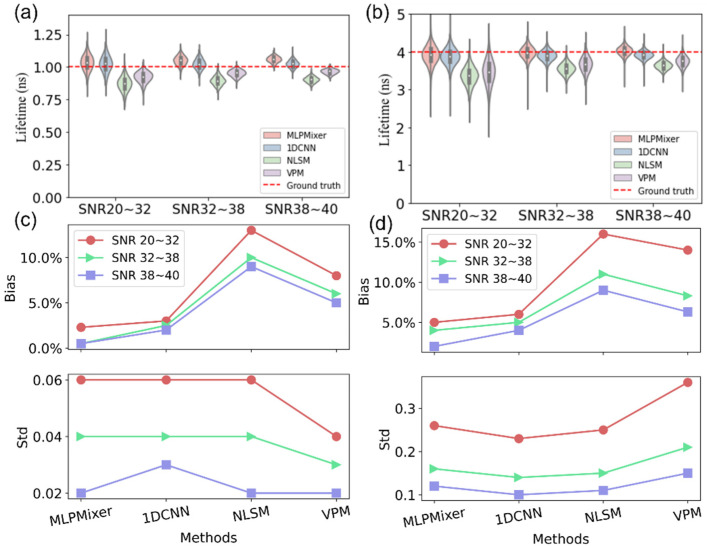
Boxplots of estimations under different signal-to-noise ratios (SNRs) with the four methods, in which the median and quartiles with whiskers reach up to 1.5 times the interquartile range. The violin plot outlines illustrate kernel probability density; for example, the width of the shaded area represents the proportion of the data. (**a**) Short lifetime (set at 1 ns) and (**b**) long lifetime (set at 4 ns) estimations from synthetic data; (**c**,**d**) corresponding bias and standard deviation plots for (**a**) short and (**b**) long lifetimes.

**Figure 4 sensors-22-07293-f004:**
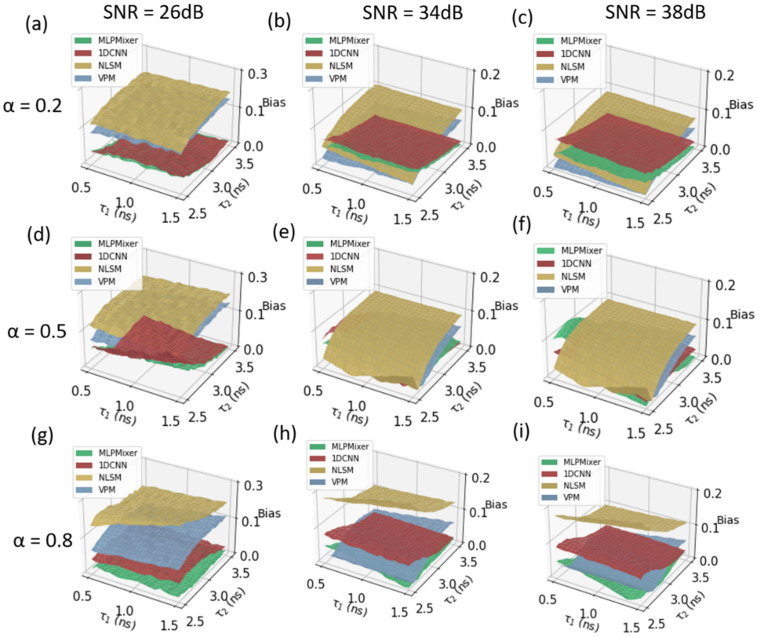
$\tau_{A}$ estimation error planes of the MLP-Mixer, 1D-CNN, NLSM, and VPM at different fraction ratios α under different signal-to-noise ratios. For signals following a biexponential decay model where $\tau_{1}\in \left(0.5,\;1.5\right)\;ns,\;\tau_{2}\epsilon \left(2.5,\;3\right)\;ns$. The error planes (**a**,**d**,**g**) SNR 26 dB, (**b**,**e**,**h**) SNR 34 dB, (**c**,**f**,**i**) SNR 38 dB, and α = 0.2, 0.5, 0.8.

**Figure 5 sensors-22-07293-f005:**
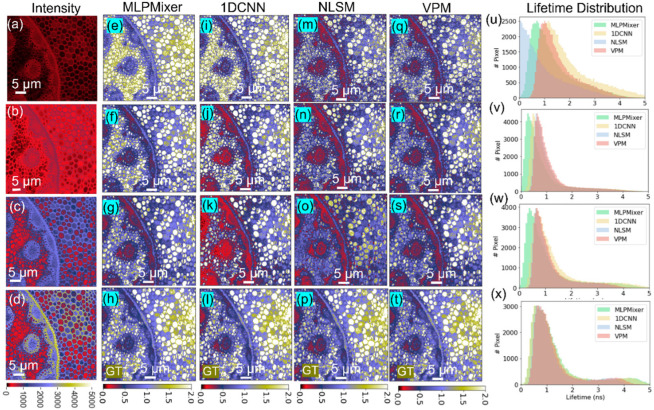
Two-photon FLIM experiments of *Convallaria majalis* cells. (**a**–**d**) Intensity images under different SNRs (3 s, 15 s, 30 s, and 90 s acquisition time). Lifetime images recovered from (**e**–**h**) the MLP-Mixer, (**i**–**l**) 1D-CNN, (**m**–**p**) NLSM, and (**q**–**t**) VPM. (**u**–**x**) Lifetime histograms for each case.

**Table 1 sensors-22-07293-t001:** Main characteristics of the MLP-Mixer and 1D-CNN lifetime processor.

Algorithm	Total Parameters	FLOPs	Total Training Time (s)	Inference Time
MLP-Mixer	61,306	5.9 B	1871.0	12.54 s
1D-CNN	17,409	5.4 B	2270.65	34.06 s

**Table 2 sensors-22-07293-t002:** Rhodamine B and 6G lifetimes with the MLP-Mixer, 1D-CNN, NLSM, and VPM.

R**hodamine B**		**SNR 26~32**	**SNR 32~38**	**SNR 38~44**	**τ_REF (ns)**
MLP-Mixer	$\tau_{mean}$ (ns)	1.70	1.73	1.71	1.72 [37]
	Std	0.40	0.46	0.50	
	Bias	1.16%	0.58%	0.58%	
	$\tau_{mean}$ (ns)	1.68	1.65	1.69	
1D-CNN	Std	0.59	0.54	0.52	
	Bias	2.33%	4.07%	1.74%	
	$\tau_{mean}$ (ns)	1.55	1.61	1.63	
NLSM	Std	0.39	0.24	0.27	
	Bias	9.88%	6.40%	5.32%	
	$\tau_{mean}$ (ns)	1.62	1.64	1.67	
VPM	Std	0.52	0.47	0.45	
	Bias	5.81%	4.56%	2.96%	
**Rhodamine 6G**		**SNR 26~32**	**SNR 32~38**	**SNR 38~44**	**τ_REF (ns)**
MLP-Mixer	$\tau_{mean}$ (ns)	4.00	4.04	4.07	4.06 [38]
	Std	0.59	0.46	0.62	
	Bias	1.48%	0.5%	0.25%	
	$\tau_{mean}$ (ns)	4.22	3.93	3.94	
1D-CNN	Std	0.78	0.7	0.72	
	Bias	3.94%	3.20%	2.95%	
	$\tau_{mean}$ (ns)	3.58	3.68	3.75	
NLSM	Std	0.53	0.27	0.34	
	Bias	11.82%	9.36%	7.64%	
	$\tau_{mean}$ (ns)	3.79	3.89	3.92	
VPM	Std	0.87	0.61	0.48	
	Bias	6.65%	4.19%	3.45%	

**Table 3 sensors-22-07293-t003:** SSIM values under different photon counts in Figure 5.

Data	MLP-Mixer	1D-CNN	NLSM	VPM
HPC	0.98	0.95	0.94	0.95
MPC	0.96	0.94	0.91	0.94
LPC	0.95	0.93	0.81	0.93

## Data Availability

Data underlying the results presented in this paper are not publicly available but may be obtained from the authors upon reasonable request.

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
