# Peer review of "Simple and Robust Deep Learning Approach for Fast Fluorescence Lifetime Imaging"

_sensors, 2022, doi:10.3390/s22197293_

Round 1

Reviewer 1 Report

Reviewer’s comments on manuscript „Simple and robust deep learning approach for fast fluorescence lifetime imaging (Sensors#1919818)” by Q. Wang et al.

The manuscript is about a deep learning approach of fluorescence lifetime measurement on cells. The Multi-layer Perceptrons-based mixer (MLPMixer) algorithm is implemented, and compared with 3 other deep learning algorithms, the 1DCNN, NLSM, and VPM. Deep learning based image analysis is a novel methodology direction aiming at high speed analyses of huge data sets (image stacks). This approach might facilitate the application of fluorescence lifetime imaging e.g. in the daily medical diagnostic practice, or in the the quick quality control of food samples. In the function of the signal-per-noise ratio (SNR), while for low SNRs MLPMixer has been found to be comparable to 1DCNN, and superior to NLSM and VPM, for large SNRs all four approaches turned to be c.a. equally efficient.

          The computational algorithms are well done, with the necessary controls. My critiques aim mainly at the way of presentation.   

Subjectal concerns:

1.      Because this approach of lifetime imaging is fairly new, the terminology is not generally known, especially for those not experts of informatics. Suggested is a leading scheme (e.g. a block diagram) in the Introduction on the general logic of deep learning algorithms, and then another scheme where the MLP-Mixer algorithm is compared with the 1D-CNN, NLSM and VPM algorithms e.g. in block diagrams. Understanding the algorithms right at the beginning is vital for the proper evaluation of the results, e.g. for judging the novelty of the Mixer algorithm compared to the others.

2.      Lines 26-28 stressing importance of lifetime detection, should be more precise. Suggested is the following sentence modification „Fluorescence lifetimes are local properties of fluorophores depending only on the physico-chemical state of the local microenvironment (e.g. pH, ionic strength) and are free of artifacts due to fluctuations in laser power, optical path and fluorophore concentration.

Intrinsic properties are factors determining quantum yield from inside the fluorophore structure. Intrinsic property is e.g. the fluorescence rate constant (kf), the reciprocal of the intrinsic fluorescence lifetime tf (kf=1/tf), depending only on the magnitude of dipole moment induced by photon absorption. But fluorescence lifetime t (and ultimately quantum yield Y=kf*t)  is determined also by the nonradiative rate constant (knr) depending on the concentration of nearby energy quenchers such as water molecules, and/or depending further on the presence of possible FRET acceptors, with rate constant kt, according to t=1/(kf+knr+kt). The rate constant kf is intrinsic, and knr and kt, are extrinsic (i.e. micro-environment dependent).

3.      In Bayesian estimation prior knowledge is also incorporated in the estimation process, like here in the process of training. What is the crucial difference between Bayesian estimation and the deep learning estimation processes? The difference, if any, should be stated in the introduction of the deep learning procedures right at the beginning.

4.      What is the cause of the similarly good efficiencies of the Mixer and 1D-CNN algorithms, independently of the SNR? And what is the reason for the little lower efficiencies of the NLSM and VPM at lower SNRs? The Discussion part should be completed with the discussion of these 2 issues.

5.      In Line 208, please state the type (name) of the 2-photon FLIM system, with the name of the manufacturer.

6.      According to Figure 2, all lifetimes are underestimated by the very nature of the TCSPC method, namely that the first photon is detected, with a modification depending on 1st, the absolute lifetime values (larger lifetimes are more reduced than smaller ones, due to larger deviations from the mean, because for exponential distribution mean=standard deviation), 2nd, the SNR (smaller reductions at larger SNRs),  and 3rd, the actual type of the implemented deep learning procedure.

7.      Can the evolution of the estimated lifetime values by the ellapsing time be visualized grafically (say graph-theoretically) in the course of these deep learning (neural network) algorithms?

In Table 2, please give also the unit (nsec) of t_REF.

Reviewer 2 Report

A fine paper - that really does what it says it does - generates lifetime images  faster (and with better contrast/quality) than existing methods. I would perhaps high-light the two photon side a little more as this is a little hidden away - but this is very minor. 

The ability to determine rapid lifetime measurements from the data. This is usually very slow and computational intensive (a real problem for images and even more so for videos). Thus it can take 5-10 mins to get a lifetime image from a microscope. This helps solve this problem.  

The topic is highly relevant in the field- Fluorescent lifetime adds the third dimension to fluorescence analysis. Traditionally Intensity vrs Wavelength - now Intensity vrs Wavelength vrs Lifetime - much richer data set.

Compare with other published material, it is a better/more rapid way of rapidly determining lifetimes - that can be used to then rapidly generate images.

The conclusions fitted well with the evidence and arguments
presented and were we’ll argued and describedAnd the references are appropriate.

Reviewer 3 Report

In this work, the deep learning (DL) algorithm of the mixer based on multilayer perceptrons (MLP mixer) (FLIM-MLP Mixer) was proposed. The FLIM-MLP mixer presents a superior performance in terms of accuracy and calculation speed obtained, and the results indicate that the proposed method can be used accurately in estimating life parameters from measured fluorescence histograms and several potential applications of FLIM.

The subject of the manuscript is interesting and can be considered for publication in Sensors after minor modifications described below: - Figure 1. The Y axis of Fig 1 (b) has a typographical error. The p.c. unit must be defined. The analysis of Fig. 1(b) should be discussed in the text. - Some equations are presented in the text without references. - Std in Table 2 must be defined. Uncertainties of physical quantities should be presented with uncertainties with appropriate rounding.
